# Tomographic Imaging of Mucociliary Clearance Following Maxillary Sinus Augmentation: A Case Series

**DOI:** 10.3390/medicina58050672

**Published:** 2022-05-18

**Authors:** Won-Bae Park, Nam-Jun Cho, Philip Kang

**Affiliations:** 1Department of Periodontology, School of Dentistry, Kyung Hee University, Seoul 02771, Korea; njkysh@naver.com; 2Division of Periodontics, Section of Oral, Diagnostic and Rehabilitation Sciences, Columbia University College of Dental Medicine, New York, NY 10032, USA; nc2881@cumc.columbia.edu

**Keywords:** mucociliary clearance, maxillary sinus augmentation, guided bone regeneration, cone-beam computed tomography, Schneiderian membrane

## Abstract

Mucociliary clearance (MCC) allows ventilation of graft particles that are displaced through a perforated Schneiderian membrane during maxillary sinus augmentation (MSA). However, it is very rarely confirmed by cone-beam computed tomographic (CBCT) images. It is not yet known how long the dislodged bone graft particles remain in the maxillary sinus or how quickly they are ventilated after MSA. The purpose of these case reports is to introduce tomographic imaging of ventilation of bone graft particles displaced through a perforated Schneiderian membrane after MSA. Four patients, who needed implant placement in the posterior maxilla, received MSA, during which the Schneiderian membrane was perforated but was not repaired. Therefore, some bone graft particles were dislocated into the sinus cavity. The sizes of the perforated membranes were measured and recorded. CBCT scans were taken at multiple time points after the surgery to visualize and trace the ejected material. In addition, the time from when the bone graft substitute was delivered to the sinus until the CBCT scans were taken was recorded. The expelled bone graft particles migrated to the ostium along the sinus wall immediately after MSA on CBCT images taken immediately after the surgery. No displaced graft particles were observed in the maxillary sinus on CBCT scans after 1 week. The CBCT scans at 6 months showed no unusual radiographic images. Within the limitations of the case reports, tomographic imaging revealed an MCC system that allows displaced graft particles to be ventilated into the ostium very early during MSA healing and not stagnate in the maxillary sinus.

## 1. Introduction

Maxillary sinus augmentation (MSA) is a procedure that enables implant placement in patients who are unable to receive conventional implant placement in the posterior maxilla due to severe pneumatization of the maxillary sinus or extreme atrophy of the maxilla [1,2,3]. The clinical and radiological outcomes of the procedure are excellent and the use of MSA is increasing gradually [3]. Postoperative complications that may occur after MSA have been studied extensively, and the most common intraoperative complication is Schneiderian membrane perforation [4,5].

Sinus floor elevation can be successfully performed when the physiology of the maxillary sinus is well maintained [6]. For this, mucociliary clearance (MCC) and ostium patency should be maintained within the normal limits [7]. MCC is the ventilation of secretory products and foreign bodies out of the sinuses. It is dependent on proper mucous viscosity and the ciliary function of the pseudostratified ciliated columnar epithelium lining the inside of the Schneiderian membrane [8].

The perforation of the Schneiderian membrane has been known to exert an adverse effect on MCC, since the mucous membrane is thickened and the ostium patency is invaded [4,5,9]. Schneiderian membrane perforation has been reported to increase the incidence of maxillary sinusitis and affect the survival rate of the implant [10,11,12]. Furthermore, displaced bone graft substitutes may cause dysventilation due to bacterial colonization or ostium plugging [13,14]. It is a common consensus that the perforated membrane should be repaired [15,16]. Therefore, various repair methods have been introduced [1]. 

Some conflicting results have been also reported. In these studies, long-term complications and implant survival rates were not affected [7,17]. This suggests that the bone graft particles dislocated through the perforated Schneiderian membrane were ventilated without stagnation in the maxillary sinus. MCC plays a role in maintaining maxillary sinus health and reducing the complications of MSA.

Unlike the nasal cavity, little is known about the time and rate of MCC in the maxillary sinus. MCC of the maxillary sinus can be confirmed only through nasal endoscopy and tomographic imaging, and is usually performed on patients with chronic or recurrent sinusitis [18,19]. However, nasal endoscopy is not possible in nonpathologic sinus conditions because it can only be performed through a surgically expanded maxillary ostium. To the authors’ knowledge, there have been no MCC studies in patients with healthy maxillary sinuses. Therefore, checking the migration of displaced graft particles after MSA using tomographic images is one of the effective ways to confirm MCC in healthy patients. In the presented case reports, the MCC of expelled bone graft particles inside the maxillary sinus is described with the use of tomographic imaging.

## 2. Presentation of Cases

Lateral MSA was performed on four patients. The Schneiderian membranes were accidentally perforated during all four procedures, and some bone graft particles were displaced through the perforation. Osteoconductive biphasic calcium phosphate (Osteon III; 0.5–10 mm particles; Genoss, Suwon, Korea) was used in all cases as the bone graft substitute. Postoperatively, all patients were covered with systemic antibiotics (Ciprofloxacin 500 mg, Ildong Pharmaceutical Co., Seoul, Korea) and nonsteroidal anti-inflammatory drugs (Etodol^®^ 200 mg, Yuhan Co., Seoul, Korea) three times a day for 14 days. Patients were also advised to rinse the mouth with 0.12% chlorhexidine solution (Hexamedine, Bukwang Pharmaceutical, Seoul, Korea) for 30 s, twice a day, for 1 week and were asked not to blow their noses. Two patients were treated with delayed implant placements six months after MSA, and the other two patients were treated with a simultaneous approach. CBCT scans were performed immediately after surgery, 7 days after surgery, and after prosthesis delivery or six months after surgery. The sizes of the membrane perforations were measured using a periodontal probe, and the time from when the bone graft substitute was delivered to the maxillary sinus until the CBCT was taken was also recorded (Table 1).

## 3. Case 1

The patient was a 62-year-old female non-smoker. Left MSA and implant placement were performed simultaneously. The size of the accidental Schneiderian membrane perforation was about 4.0 mm, and repair was not performed. To prevent displacement of the bone graft particles, condensation was carried out in the direction of the sinus floor, and overfilling was avoided. Two implants (Implantium 4.3 × 10, Dentium, Suwon, Korea) were placed in the area of #26 and #27 missing teeth. The time from delivery of bone graft substitute into the elevated maxillary sinus to CBCT imaging immediately after surgery was approximately 30 min (Table 1). There were transient nasal bleeding and facial swelling after the operation but no other postoperative complications. A coronal image of CBCT before the surgery at site #26 showed that the ostium was patent and the Schneiderian membrane was slightly thickened (Figure 1a). In the coronal image of CBCT immediately after the surgery, a mass of displaced graft particles was observed at the entrance of the ostium (white arrow in Figure 1b). In the sagittal image of the CBCT taken immediately after the surgery, graft particles at the entrance to the ostium (white arrow) and graft particles passing through the infundibulum (yellow arrow) were observed (Figure 1c). An axial image of the CBCT taken immediately after the surgery showed that the graft particles passed through the infundibulum and were ventilated into the middle meatus (yellow arrow in Figure 1d). The ejected bone graft particles were not observed on the CBCT scan 1 week after the surgery (Figure 1e). CBCT scan at prosthesis delivery showed a decrease in the membrane thickening (Figure 1f).

## 4. Case 2 

The patient was a 36-year-old male smoker. Lateral sinus floor elevation and implant placement were simultaneously performed in both pneumatized maxillary sinuses. There was no history of preoperative sinonasal treatment, and the mucosal thickness of the maxillary sinus was very thin. During MSA, there was a wide perforation of the membrane in the left (10 mm) and right (15 mm) sinuses. Since membrane repair was not performed, the bone graft substitute had to be exposed in the perforated area. The bone graft material was filled only on the sinus floor to prevent it from spreading into the maxillary sinus as much as possible. The implants placed were 4.8 × 10 mm Implantium (Dentium, Suwon, Korea). The time elapsed between the delivery of the bone graft material and postoperative CBCT imaging was 60 min for the right surgical site and 30 min for the left surgical site (Table 1). On the CBCT scan taken immediately after the surgery, saline and bone graft particles leaking through the perforated Schneiderian membrane were observed (Figure 2a–f). Bone graft particles were floating in physiological saline, some with individual particles and others in lumps. The displaced bone graft particles were observed to migrate towards the natural ostium along the medial and lateral sinus wall (Figure 2b–d,f). After surgery, the patient had some nasal bleeding for 2–3 days but no other complications. Six months after the operation, the final prosthesis was delivered. There was no expelled bone graft particle in the right and left maxillary sinus and no membrane thickening on CBCT taken 6 months after surgery (Figure 2g–l).

## 5. Case 3

The patient was a 45-year-old male non-smoker. Implant placement was planned as a delayed approach to be installed 6 months after MSA in the left pneumatized maxillary sinus. The Schneiderian membrane was accidentally perforated and not repaired. The size of the membrane perforation was about 6 mm (Table 1). The time from the delivery of the bone graft to the immediate CBCT scan was 30 min. Coronal images of site #27 showed lumps of displaced bone graft particles floating in leaked saline (Figure 3a). The patient did not complain of other complications, except for swelling and pain for 1 week. CBCT scan taken 1 week postoperatively showed only membrane thickening and an absence of ejected bone graft particles (Figure 3b). At 6 months postoperatively, membrane thickening decreased and no other findings were found (Figure 3c).

## 6. Case 4

The last case was a 47-year-old non-smoking patient. In the left pneumatized maxillary sinus, MSA and implant placement were performed simultaneously at sites #26 and #27. During the procedure, the Schneiderian membrane was accidentally perforated (~8 mm), but was not repaired. A CBCT scan was taken about 20 min immediately after the bone graft substitute was delivered (Table 1). The patient reported postoperative pain and swelling. The immediate CBCT images showed displaced bone graft particles mixed with saline at site #26 (Figure 4a). All dislocated bone graft particles were not present on the CBCT scan 1 week after the surgery (Figure 4b). CBCT scan at 6 months postoperatively showed a decrease in membrane thickening and no ejected bone graft particles in the maxillary sinus (Figure 4c).

## 7. Results

Four cases of MSA without repair of the accidentally perforated Schneiderian membrane resulted in displacement of the bone graft particles into the maxillary sinus cavity. However, the dislocated bone graft particles rapidly migrated to the ostium along the sinus wall immediately after the surgery. Most expelled graft particles were not observed in the maxillary sinus on CBCT scans after 1 week. In the CBCT images taken after 6 months, abnormal mucosal condition was absent.

## 8. Discussion

Ventilation of bone graft particles displaced through the perforated Schneiderian membrane will not affect the healing of MSA and allows homeostatic sinus physiology to be maintained. On the other hand, dysventilation of dislocated bone graft particles may cause postoperative complications. The presented cases visualized on CBCT scans showed the displaced bone graft particles began to be ventilated by the MCC system immediately after surgery, and the clearance was completed within 1 week after the surgery. The expelled bone graft particles did not affect the success of MSA.

The physiology of the maxillary sinus is maintained by the MCC and the patency of the maxillary ostium [7]. The MCC system is an important host defense mechanism that maintains homeostasis by protecting the body from invading foreign particles, including bacteria [20]. The MCC of the maxillary sinus allows the mucus to ventilate through the ostium to the middle meatus of the nasal cavity. Small particles and microbes in the sinus cavity are mixed with mucus made by goblet cells and transported to the ostium by ciliary movement [8,21]. Although the exact rate of clearance in the normal maxillary sinus is not known due to the limitation of access, it can be extrapolated from the clearance time of the nasal cavity because they share continuous pseudostratified ciliated columnar epithelium [21]. The nasal MCC rate is usually measured by the saccharine test or radioisotope technique: the mean nasal clearance time is 7–15 min in healthy adults, and over 30 min is judged abnormal [22]. The clearance rate is about 10 mm/min in the posterior part of the nasal cavity [22]. Theoretically, it does not take 30 min for displaced graft particles to migrate from the sinus floor to the ostium, but it can change depending on the function and density of the cilia, mucus quality, the position of the maxillary sinus, and the presence of sinus disease [23]. In case 1 and case 2, CBCT images taken immediately after the surgery, the expelled graft particles were observed to migrate very early into the ostium, pass through the infundibulum and ventilate into the middle meatus. The time from MSA to tomographic scan is usually 20–30 min, and as in case 2, bilateral MSA takes as long as 30–60 min.

Perforation of the Schneiderian membrane during MSA has been known to provide various complications [4,5,9,24], and it can adversely affect the MCC and patency of maxillary ostium [7]. Therefore, repair of the perforated Schneiderian membrane is recommended to reduce postoperative complications and increase the success of MSA [25]. The repair procedure does not affect the lateral sinus floor elevation and implant survival rate [1]. Several repair methods have been introduced according to various perforation sizes, but they are technique sensitive and difficult to achieve [16,26]. In general, displaced bone graft substitute inhibits the ventilation by obstructing the ostium or colonizes with resident bacteria in the maxillary sinus, causing mucosal thickening [14]. There is also a report of displaced graft material plugging the ostium, resulting in postoperative maxillary sinusitis [13]. However, Park et al. reported that MSA can be successful even without repairing the perforated Schneiderian membrane [27]. In the present study, the perforations were not repaired. Therefore, some of the bone graft particles were displaced into the maxillary sinus. We suggested that the ostium plugging of graft particles, sometimes found in postoperative maxillary sinusitis, blocks the infundibulum by the thickened Schneiderian membrane, but the bone graft particles were not actually caught at the ostium. In other words, ostium plugging is a secondary phenomenon, not a primary cause.

It is impossible for a nasal endoscope to approach the healthy maxillary sinus. Surgical expansion of the anatomical ostium allows entry of the nasal endoscope into the maxillary sinus. Therefore, it is difficult to identify displaced bone graft material with a nasal endoscope in MSA cases of healthy sinus patients. CBCT is an alternative to confirm the ventilation of displaced graft particles. CBCT imaging after MSA is usually performed preoperatively, either immediately after the surgery or after final prosthesis delivery. However, CBCT is not commonly taken 1 week postoperatively. Thus, displaced bone graft particles were misinterpreted as being left in the sinus cavity for a long time without ventilating. CBCT after 1 week of surgery is recommended if any displacement of bone graft particles is suspected, irrespective of the repair of perforated Schneiderian membrane.

In the present patients, the ventilation of ejected bone graft particles began to be detected on a CBCT scan taken immediately after the operation, and the expelled material did not exist in the sinus cavity by the second week. This suggests that displaced bone graft particles are well ventilated to the middle meatus within one week. The dislocated bone graft did not stay in the maxillary sinus enough to colonize the resident bacteria. In addition, the lumps of graft material did not obstruct the anatomical ostium and did not cause severe postoperative thickening of the Schneiderian membrane. 

Nonetheless, some limitations of these case reports should be noted. The limited number of cases contributes to a weaker body of evidence and might have introduced bias in the observations. Moreover, the exact MCC time and rate could not be known from the tomographic imaging alone. For future studies, a robust clinical guideline may be presented if many clinicians are involved and many case numbers are secured.

## 9. Conclusions

Within the limitations of the case reports, tomographic imaging showed that bone graft particles displaced into the perforated Schneiderian membrane during maxillary sinus augmentation were successfully ventilated by mucociliary clearance very early within the first week of healing. The expelled bone graft material was no longer stagnant in the maxillary sinus and did not cause any adverse postoperative complications.

## Figures and Tables

**Figure 1 medicina-58-00672-f001:**
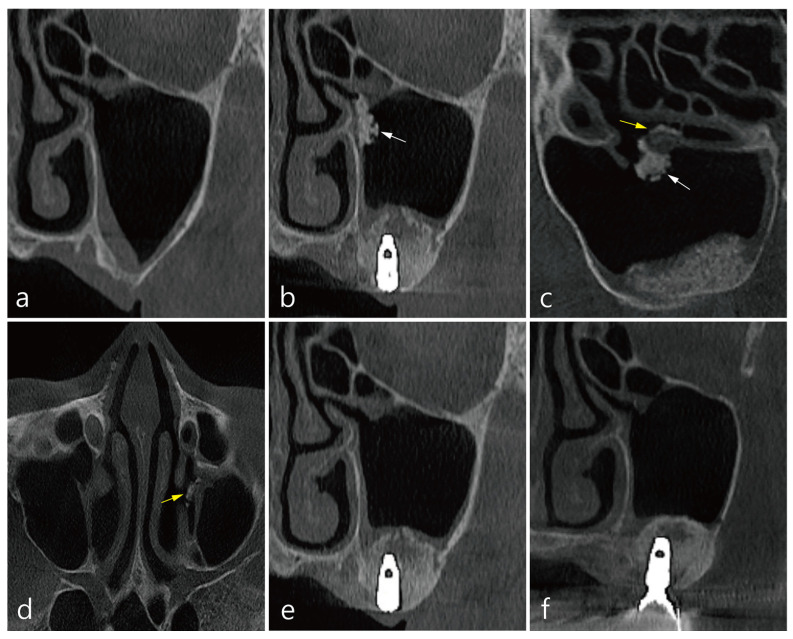
CBCT images of mucociliary clearance (MCC) in case 1: (**a**) A coronal image of the left maxillary sinus before MSA. The ostium was open and slight membrane thickening was observed. (**b**) In the coronal image of CBCT immediately after the surgery, a mass of displaced graft particles was observed (white arrow). (**c**) In the sagittal image, graft particles at the entrance to the ostium (white arrow) and particles passing through the infundibulum (yellow arrow) were observed. (**d**) The axial image showed that graft particles ventilated into the middle meatus (yellow arrow). (**e**) The expelled particles were not observed on CBCT scan 1 week after the surgery. (**f**) The CBCT scan at prosthesis delivery showed a decrease in membrane thickening.

**Figure 2 medicina-58-00672-f002:**
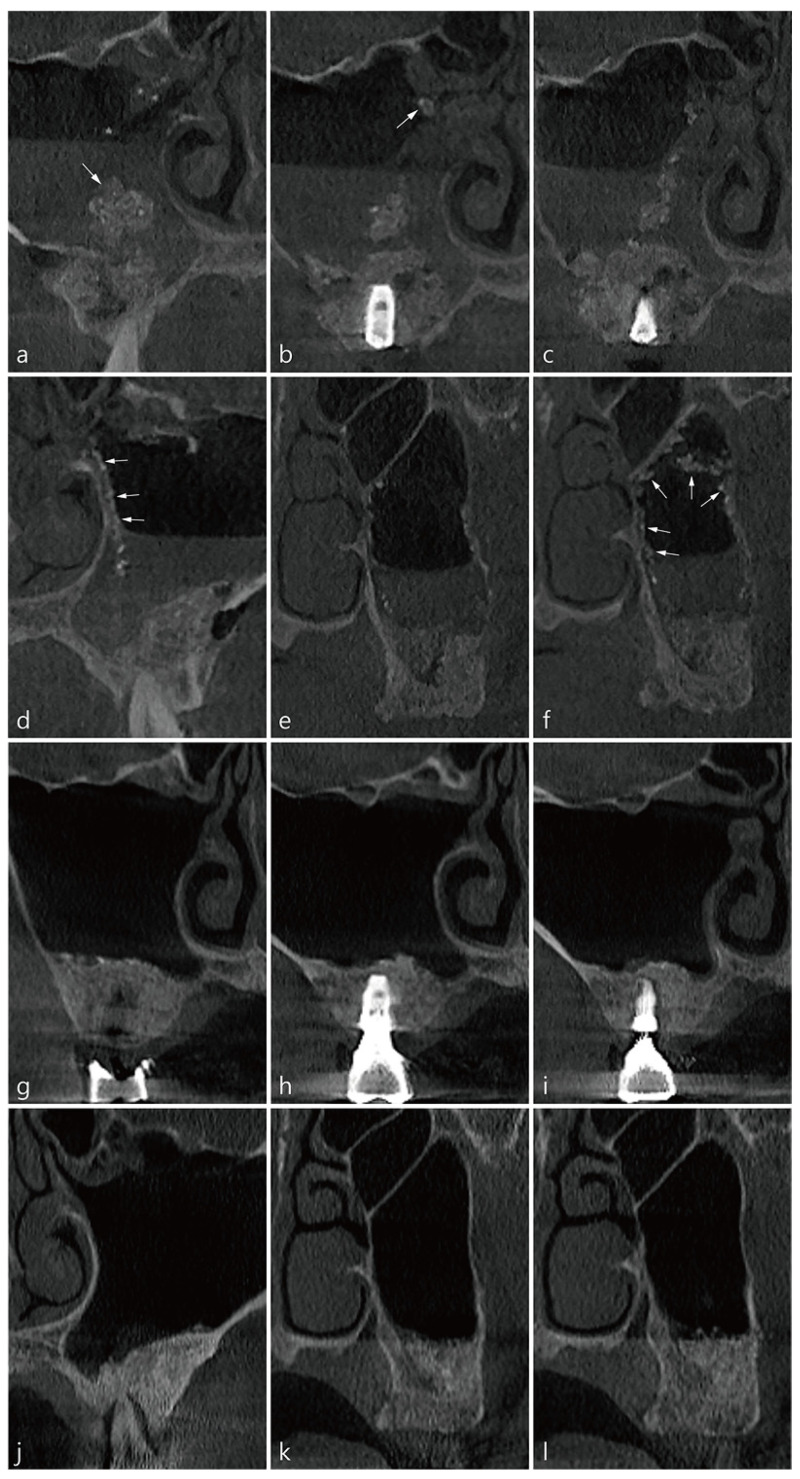
CBCT images of MCC in case 2: (**a**) In the coronal image of tooth #15, a large mass of bone graft material floating in saline (white arrow) and some particles near the ostium in the sinus cavity were observed. (**b**) At the #16 implant site, a small lump of graft is visible at the entrance of the ostium (white arrow). (**c**) The expelled material is moving towards the ostium along the medial wall of the maxillary sinus. (**d**) In the coronal image of tooth #25, graft particles are moving towards the ostium along the medial wall of the maxillary sinus (white arrows). (**e**) Particles were observed in the medial wall and lateral wall at site #27. (**f**) Graft particles were observed along the superior, medial, and lateral walls (white arrows). (**g**–**l**) Coronal images of the CBCT scan 6 months postoperatively at the same sites as in (**a**–**f**). The ejected graft was all ventilated and not in the maxillary sinus. No mucosal thickening was observed.

**Figure 3 medicina-58-00672-f003:**
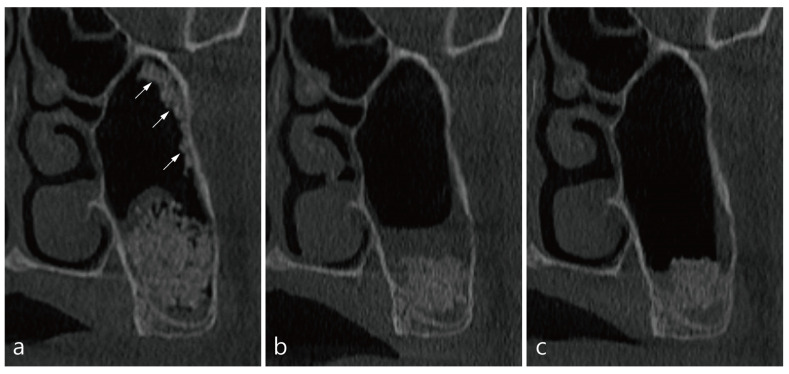
CBCT images of MCC in case 3: (**a**) The coronal image of the CBCT immediately after the surgery at site #27 showed displaced bone graft particles along the lateral wall of the maxillary sinus (white arrows). (**b**) No expelled particles were found in the CBCT after 1 week of operation, and only mild membrane thickening was observed. (**c**) The membrane thickening decreased in CBCT after 6 months postoperatively.

**Figure 4 medicina-58-00672-f004:**
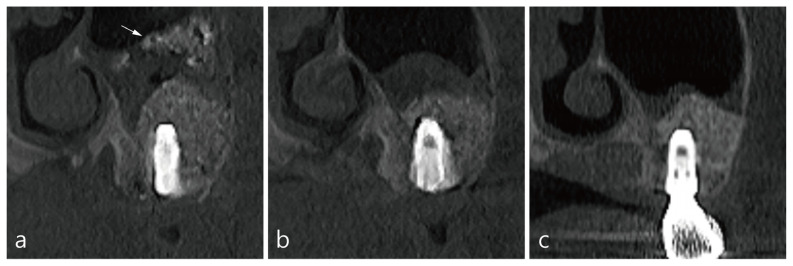
CBCT images of MCC in case 4: (**a**) The coronal image of #26 implant site on CBCT taken immediately after the operation showed a lump of bone graft particles in leaking saline (white arrow). (**b**) CBCT scan taken 1 week after the surgery showed only membrane thickening and no displaced bone graft particles. (**c**) At 6 months postoperatively, membrane thickening decreased, and no other findings were noted.

**Table 1 medicina-58-00672-t001:** The demographics of the included patients.

Case	Sex	Age	Smoking	Implant Sites	Membrane Perforation (mm)	Membrane Repair	Time (from Graft Delivery to CBCT Scan)
1	F	62	No	#26, #27	4	No	about 30 min
2	M	36	Yes	#16, #17, #26, #27	Right: 15, Left: 10	No	Right: about 60 min Left: about 30 min
3	M	45	No	Delayed approach (Left sinus)	6	No	about 30 min
4	M	47	No	#26, #27	8	No	about 20 min

Note: Cone-beam computed tomography (CBCT).

## Data Availability

All data and material are presented in the manuscript.

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
