# Peer review of "Tomographic Imaging of Mucociliary Clearance Following Maxillary Sinus Augmentation: A Case Series"

_medicina, 2022, doi:10.3390/medicina58050672_

Round 1

Reviewer 1 Report

The title could be modified writing “A case series” instead of “Case reports”.

Osteon III characteristics, in terms of composition and particles dimension, should be reported.

Why implants images are not showed in Case 3?

Which instructions patients’ received after the surgery, in terms of drugs and other aspects

Author Response

Dear reviewer,

Thank you very much for your suggestions.  My responses to your comments are below, and the changes are highlighted in the main text.

The title could be modified writing “A case series” instead of “Case reports”.

Response: The title has been changed.

Osteon III characteristics, in terms of composition and particles dimension, should be reported.

Response: The requested information has been added to the text.

Why implants images are not showed in Case 3?

Response: This was a delayed approach, and implants have not been placed.

Which instructions patients’ received after the surgery, in terms of drugs and other aspects

Response: The post-op instructions given to all patients have been added.

Reviewer 2 Report

The case reports, entitled “Tomographic Imaging of Mucociliary Clearance Following Maxillary Sinus Augmentation: Case Reports”, describe four cases of the mucociliary clearance of bone graft particles displaced through a perforated Schneiderian membrane after maxillary sinus augmentation using CBCT. The topic of the present case reports is novel and very likely to draw the attention of interested dental practitioners. The reviewer only has two suggestions as follows:

  1. Could the authors provide some clinical images showing the Schneiderian membrane perforation?
  2. Please emphasize in the “Case Presentation” section that the Schneiderian membranes were perforated accidentally not intentionally, and no collagen membranes were used.

Author Response

Dear reviewer,

Thank you very much for your suggestions. My responses to your comments are below, and the changes are highlighted in the main text.

Could the authors provide some clinical images showing the Schneiderian membrane perforation?

Response: Your suggestion is extremely valid.  However, at the time of each surgery, photographs were not obtained although perforations were visible.  The dimension of perforation is described in the text and further verified with CBCT images with displaced bone grafts in the sinus.

Please emphasize in the “Case Presentation” section that the Schneiderian membranes were perforated accidentally not intentionally, and no collagen membranes were used.

Response: Thank you for the suggestion.  The word "accidental" has been added to each case description.